# A Graphical Transformation for Belief Propagation: Maximum Weight Matchings and Odd-Sized Cycles

**Jinwoo Shin**
Department of Electrical Engineering
Korea Advanced Institute of Science and Technology
Daejeon, 305-701, Republic of Korea
jinwoos@kaist.ac.kr

**Andrew E. Gelfand** *
Department of Computer Science
University of California, Irvine
Irvine, CA 92697-3435, USA
agelfand@ics.uci.edu

**Michael Chertkov**
Theoretical Division &
Center for Nonlinear Studies
Los Alamos National Laboratory
Los Alamos, NM 87545, USA
chertkov@lanl.gov

## Abstract

Max-product 'belief propagation' (BP) is a popular distributed heuristic for finding the Maximum A Posteriori (MAP) assignment in a joint probability distribution represented by a Graphical Model (GM). It was recently shown that BP converges to the correct MAP assignment for a class of loopy GMs with the following common feature: the Linear Programming (LP) relaxation to the MAP problem is tight (has no integrality gap). Unfortunately, tightness of the LP relaxation does not, in general, guarantee convergence and correctness of the BP algorithm. The failure of BP in such cases motivates reverse engineering a solution – namely, given a tight LP, can we design a 'good' BP algorithm.

In this paper, we design a BP algorithm for the Maximum Weight Matching (MWM) problem over general graphs. We prove that the algorithm converges to the correct optimum if the respective LP relaxation, which may include inequalities associated with non-intersecting odd-sized cycles, is tight. The most significant part of our approach is the introduction of a novel graph transformation designed to force convergence of BP. Our theoretical result suggests an efficient BP-based heuristic for the MWM problem, which consists of making sequential, "cutting plane", modifications to the underlying GM. Our experiments show that this heuristic performs as well as traditional cutting-plane algorithms using LP solvers on MWM problems.

## 1 Introduction

Graphical Models (GMs) provide a useful representation for reasoning in a range of scientific fields [1, 2, 3, 4]. Such models use a graph structure to encode the joint probability distribution, where vertices correspond to random variables and edges (or lack of thereof) specify conditional dependencies. An important inference task in many applications involving GMs is to find the most likely assignment to the variables in a GM - the maximum a posteriori (MAP) configuration. Belief Propagation (BP) is a popular algorithm for approximately solving the MAP inference problem. BP is an iterative, message passing algorithm that is exact on tree structured GMs. However, BP often shows remarkably strong heuristic performance beyond trees, i.e. on GMs with loops. Distributed implementation, associated ease of programming and strong parallelization potential are among the main reasons for the popularity of the BP algorithm, e.g., see the parallel implementations of [5, 6].

The convergence and correctness of BP was recently established for a certain class of loopy GM formulations of several classic combinatorial optimization problems, including matchings [7, 8, 9], perfect matchings [10], independent sets [11] and network flows [12]. The important common

feature of these instances is that BP converges to a correct MAP assignment when the Linear Programming (LP) relaxation of the MAP inference problem is tight, i.e., it shows no integrality gap. While this demonstrates that LP tightness is necessary for the convergence and correctness of BP, it is unfortunately not sufficient in general. In other words, BP may not work even when the corresponding LP relaxation to the MAP inference problem is tight. This motivates a quest for improving BP-based MAP solvers so that they work when the LP is tight.

In this paper, we consider a specific class of GMs corresponding to the Maximum Weight Matching (MWM) problem and study if BP can be used as an iterative, message passing-based LP solver when the MWM LP (relaxation) is tight. It was recently shown [15] that a MWM can be found in polynomial time by solving a carefully chosen sequence of LP relaxations, where the sequence of LPs are formed by adding and removing sets of so-called "blossom" inequalities [13] to the base LP relaxation. Utilizing successive LP relaxations to solve the MWM problem is an example of the popular cutting plane method for solving combinatorial optimization problems [14]. While the approach in [15] is remarkable in that one needs only a polynomial number of "cut" inequalities, it unfortunately requires solving an emerging sequence of LPs via traditional, centralized methods (e.g., ellipsoid, interior-point or simplex) that may not be practical for large-scale problems. This motivates our search for an efficient and distributed BP-based LP solver for this class of problems.

Our work builds upon that of Sanghavi, Malioutov and Willsky [8], who studied BP for the GM formulation of the MWM problem on an arbitrary graph. The authors showed that max-product BP converges to the correct, MAP solution if the base LP relaxation with no blossom - referred to herein as MWM-LP - is tight. Unfortunately, the tightness is not guaranteed in general, and the convergence and correctness for max-product BP do not readily extend to a GM with blossom constraints.

To resolve this issue, we propose a novel GM formulation of the MWM problem and show that max-product BP on this new GM converges to the MWM assignment as long as the MWM-LP relaxation with blossom constraints is tight. The only restriction placed on our GM construction is that the set of blossom constraints added to the base MWM-LP be non-intersecting (in edges). Our GM construction is motivated by the so-called 'degree-two' (DT) condition, which requires that every variable in the GM be associated to at most two factor functions. The DT condition is necessary for analysis of BP using the computational tree technique, developed and advanced in [7, 8, 12, 16, 18, 19]. Note, that the DT condition is not satisfied by the standard MWM GM formulation, and hence, we design a new GM that satisfies the DT condition via a clever graphical transformation - namely, collapsing odd-sized cycles and defining new weights on the contracted graph. Importantly, the MAP assignments of the two GMs are in one-to-one correspondence guaranteeing that a solution to the original problem can be recovered.

Our theoretical result suggests a cutting-plane approach to the MWM problem, where BP is used as the LP solver. In particular, we examine the BP solution to identify odd-sized cycle constraints - "cuts" - to add to the MWM-LP relaxation; then construct a new GM using our graphical transformation, run BP and repeat. We evaluate this heuristic empirically and show that its performance is close to a traditional cutting-plane approach employing an LP solver rather than BP. Finally, we note that the DT condition may neither be sufficient nor necessary for BP to work. It was necessary, however, to provide theoretical guarantees for the special class of GMs considered. To our knowledge, our result is the first to suggest how to "fix" BP via a graph transformation so that it works properly, i.e., recovers the desired LP solution. We believe that our success in crafting a graphical transformation will offer useful insight into the design and analysis of BP algorithms for a wider class of problems.

**Organization.** In Section 2, we introduce a standard GM formulation of the MWM problem as well as the corresponding BP and LP. In Section 3, we introduce our new GM and describe performance guarantees of the respective BP algorithm. In Section 4, we describe a cutting-plane(-like) method using BP for the MWM problem and show its empirical performance for random MWM instances.

## 2 Preliminaries

### 2.1 Graphical Model for Maximum Weight Matchings

A joint distribution of $n$ (discrete) random variables $Z = [Z_i] \in \Omega^n$ is called a Graphical Model (GM) if it factorizes as follows: for $z = [z_i] \in \Omega^n$,

$$\Pr[Z = z] \; \propto \; \prod\nolimits_{\alpha \in F} \psi_\alpha(z_\alpha), \tag{1}$$

where $F$ is a collection of subsets of $\Omega$, $z_\alpha = [z_i : i \in \alpha \subset \Omega]$ is a subset of variables, and $\psi_\alpha$ is some (given) non-negative function. The function $\psi_\alpha$ is called a factor (variable) function if $|\alpha| \geq 2$ ($|\alpha| = 1$). For variable functions $\psi_\alpha$ with $\alpha = \{i\}$, we simply write $\psi_\alpha = \psi_i$. One calls $z$ a valid assignment if $\Pr[Z = z] > 0$. The MAP assignment $z^*$ is defined as

$$z^* \; = \; \arg\max_{z \in \Omega^n} \Pr[Z = z].$$

Let us introduce the Maximum Weight Matching (MWM) problem and its related GM. Suppose we are given an undirected graph $G = (V, E)$ with weights $\{w_e : e \in E\}$ assigned to its edges. A *matching* is a set of edges without common vertices. The weight of a matching is the sum of corresponding edge weights. The MWM problem consists of finding a matching of maximum weight. Associate a binary random variable with each edge $X = [X_e] \in \{0, 1\}^{|E|}$ and consider the probability distribution: for $x = [x_e] \in \{0, 1\}^{|E|}$,

$$\Pr[X = x] \; \propto \; \prod_{e \in E} e^{w_e x_e} \prod_{i \in V} \psi_i(x) \prod_{C \in \mathcal{C}} \psi_C(x), \tag{2}$$

where

$$\psi_i(x) = \begin{cases} 1 & \text{if } \sum_{e \in \delta(i)} x_e \leq 1 \\ 0 & \text{otherwise} \end{cases} \qquad \text{and} \qquad \psi_C(x) = \begin{cases} 1 & \text{if } \sum_{e \in E(C)} x_e \leq \frac{|C|-1}{2} \\ 0 & \text{otherwise} \end{cases}.$$

Here $\mathcal{C}$ is a set of odd-sized cycles $C \subset 2^V$, $\delta(i) = \{(i, j) \in E\}$ and $E(C) = \{(i, j) \in E : i, j \in C\}$. Throughout the manuscript, we assume that cycles are non-intersecting in edges, i.e., $E(C_1) \cap E(C_2) = \emptyset$ for all $C_1, C_2 \in \mathcal{C}$. It is easy to see that a MAP assignment $x^*$ for the GM (2) induces a MWM in $G$. We also assume that the MAP assignment is unique.

### 2.2 Belief Propagation and Linear Programming for Maximum Weight Matchings

In this section, we introduce max-product Belief Propagation (BP) and the Linear Programming (LP) relaxation to computing the MAP assignment in (2). We first describe the BP algorithm for the general GM (1), then tailor the algorithm to the MWM GM (2). The BP algorithm updates the set of $2|\Omega|$ messages $\{m_{\alpha \to i}^t(z_i), m_{i \to \alpha}^t(z_i) : z_i \in \Omega\}$ between every variable $i$ and its associated factors $\alpha \in F_i = \{\alpha \in F : i \in \alpha, |\alpha| \geq 2\}$ using the following update rules:

$$m_{\alpha \to i}^{t+1}(z_i) = \sum_{z' : z_i' = z_i} \psi_\alpha(z') \prod_{j \in \alpha \setminus i} m_{j \to \alpha}^t(z_j') \qquad \text{and} \qquad m_{i \to \alpha}^{t+1}(z_i) = \psi_i(z_i) \prod_{\alpha' \in F_i \setminus \alpha} m_{\alpha' \to i}^t(z_i).$$

Here $t$ denotes time and initially $m_{\alpha \to i}^0(\cdot) = m_{i \to \alpha}^0(\cdot) = 1$. Given a set of messages $\{m_{i \to \alpha}(\cdot), m_{\alpha \to i}(\cdot))\}$, the BP (max-marginal) beliefs $\{n_i(z_i)\}$ are defined as follows:

$$n_i(z_i) \; = \; \psi_i(z_i) \prod\nolimits_{\alpha \in F_i} m_{\alpha \to i}(z_i).$$

For the GM (2), we let $n_e^t(\cdot)$ to denote the BP belief on edge $e \in E$ at time $t$. The algorithm outputs the MAP estimate at time $t$, $x^{\text{BP}}(t) = [x_e^{\text{BP}}(t)] \in [0, ?, 1]^{|E|}$, using the using the beliefs and the rule:

$$x_e^{\text{BP}}(t) = \begin{cases} 1 & \text{if } n_e^t(0) < n_e^t(1) \\ ? & \text{if } n_{ij}^t(0) = n_e^t(1) \\ 0 & \text{if } n_e^t(0) > n_e^t(1) \end{cases}.$$

The LP relaxation to the MAP problem for the GM (2) is:

$$\text{C-LP}: \qquad \max \sum_{e \in E} w_e x_e$$

$$\text{s.t.} \qquad \sum_{e \in \delta(i)} x_e \leq 1, \quad \forall i \in V, \qquad \sum_{e \in E(C)} x_e \leq \frac{|C| - 1}{2}, \quad \forall C \in \mathcal{C}, \qquad x_e \in [0, 1].$$

Observe that if the solution $x^{\text{C-LP}}$ to C-LP is integral, i.e., $x^{\text{C-LP}} \in \{0,1\}^{|E|}$, then it is a MAP assignment, i.e., $x^{\text{C-LP}} = x^*$. Sanghavi, Malioutov and Willsky [8] proved the following theorem connecting the performance of BP and C-LP in a special case:

**Theorem 2.1.** *If $\mathcal{C} = \emptyset$ and the solution of C-LP is integral and unique, then $x^{BP}(t)$ under the GM (2) converges to the MWM assignment $x^*$.*

Adding small random component to every weight guarantees the uniqueness condition required by Theorem 2.1. A natural hope is that Theorem 2.1 extends to a non-empty $\mathcal{C}$ since adding more cycles can help to reduce the integrality gap of C-LP. However, the theorem does not hold when $\mathcal{C} \neq \emptyset$. For example, BP does not converge for a triangle graph with edge weights $\{2, 1, 1\}$ and $\mathcal{C}$ consisting of the only cycle. This is true even though the solution to its C-LP is unique and integral.

## 3    A Graphical Transformation for Convergent & Correct BP

The loss of convergence and correctness of BP when the MWM LP is tight (and unique) but $\mathcal{C} \neq \emptyset$ motivates the work in this section. We resolve the issue by designing a new GM, equivalent to the original GM, such that when BP is run on this new GM it converges to the MAP/MWM assignment whenever the LP relaxation is tight and unique - even if $\mathcal{C} \neq \emptyset$. The new GM is defined on an auxiliary graph $G' = (V', E')$ with new weights $\{w'_e : e \in E'\}$, as follows:

$$V' = V \cup \{i_C : C \in \mathcal{C}\}, \qquad E' = E \cup \{(i_C, j) : j \in V(C), C \in \mathcal{C}\} \setminus \{e : e \in \cup_{C \in \mathcal{C}} E(C)\}$$

$$w'_e = \begin{cases} \frac{1}{2}\sum_{e' \in E(C)}(-1)^{d_C(j,e')}w_{e'} & \text{if } e = (i_C, j) \quad \text{for some } C \in \mathcal{C} \\ w_e & \text{otherwise} \end{cases}.$$

Here $d_C(j, e)$ is the graph distance of $j$ and $e$ in cycle $C = (j_1, j_2, \ldots, j_k)$, e.g., if $e = (j_2, j_3)$, then $d_C(j_1, e) = 1$.

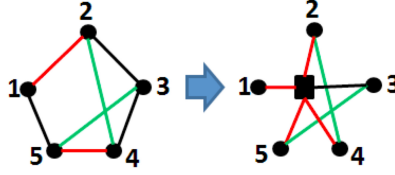

Figure 1: Example of original graph $G$ (left) and new graph $G'$ (right) after collapsing cycle $C = (1, 2, 3, 4, 5)$. In the new graph $G'$, edge weight $w_{1C} = 1/2(w_{12} - w_{23} + w_{34} - w_{45} + w_{15})$.

Associate a binary variable with each new edge and consider the new probability distribution on $y = [y_e : e \in E'] \in \{0, 1\}^{|E'|}$:

$$\Pr[Y = y] \propto \prod_{e \in E'} e^{w'_e y_e} \prod_{i \in V} \psi_i(y) \prod_{C \in \mathcal{C}} \psi_C(y), \tag{3}$$

where

$$\psi_i(y) = \begin{cases} 1 & \text{if } \sum_{e \in \delta(i)} y_e \leq 1 \\ 0 & \text{otherwise} \end{cases} \qquad \psi_C(y) = \begin{cases} 0 & \text{if } \sum_{e \in \delta(i_C)} y_e > |C| - 1 \\ 0 & \text{if } \sum_{j \in V(C)}(-1)^{d_C(j,e)}y_{i_C, j} \notin \{0, 2\} \text{ for some } e \in E(C) \\ 1 & \text{otherwise} \end{cases}.$$

It is not hard to check that the number of operations required to update messages at each round of BP under the above GM is $O(|V||E|)$, as messages updates involving factor $\psi_C$ require solving a MWM problem on a simple cycle – which can be done efficiently via dynamic programming in time $O(|C|)$ – and the summation of the numbers of edges of non-intersecting cycles is at most $|E|$. We are now ready to state the main result of this paper.

**Theorem 3.1.** *If the solution of C-LP is integral and unique, then the BP-MAP estimate $y^{BP}(t)$ under the GM (3) converges to the corresponding MAP assignment $y^*$. Furthermore, the MWM assignment $x^*$ is reconstructible from $y^*$ as:*

$$x_e^* = \begin{cases} \frac{1}{2}\sum_{j \in V(C)}(-1)^{d_C(j,e)}y_{i_C,j}^* & \text{if } e \in \bigcup_{C \in \mathcal{C}} E(C) \\ y_e^* & \text{otherwise} \end{cases}. \tag{4}$$

The proof of Theorem 3.1 is provided in the following sections. We also establish the convergence time of the BP algorithm under the GM (3) (see Lemma 3.2). We stress that the new GM (3) is designed so that each variable is associated to at most two factor nodes. We call this condition, which did not hold for the original GM (2), the 'degree-two' (DT) condition. The DT condition will play a critical role in the proof of Theorem 3.1. We further remark that even under the DT condition and given tightness/uniqueness of the LP, proving correctness and convergence of BP is still highly non trivial. In our case, it requires careful study of the computation tree induced by BP with appropriate truncations at its leaves.

## 3.1 Main Lemma for Proof of Theorem 3.1

Let us introduce the following auxiliary LP over the new graph and weights.

$$
\text{C-LP}' : \quad \max \sum_{e \in E'} w'_e y_e
$$

$$
\text{s.t.} \quad \sum_{e \in \delta(i)} y_e \le 1, \quad \forall i \in V, \quad y_e \in [0,1], \quad \forall e \in E', \tag{5}
$$

$$
\sum_{j \in V(C)} (-1)^{d_C(j,e)} y_{i_C,j} \in [0,2], \quad \forall e \in E(C), \quad \sum_{e \in \delta(i_C)} y_e \le |C| - 1, \quad \forall C \in \mathcal{C}. \tag{6}
$$

Consider the following one-to-one linear mapping between $x = [x_e : e \in E]$ and $y = [y_e : e \in E']$:

$$
y_e = \begin{cases} \sum_{e' \in E(C) \cap \delta(i)} x_{e'} & \text{if } e = (i, i_C) \\ x_e & \text{otherwise} \end{cases} \qquad x_e = \begin{cases} \frac{1}{2} \sum_{j \in V(C)} (-1)^{d_C(j,e)} y_{i_C,j} & \text{if } e \in \bigcup_{C \in \mathcal{C}} E(C) \\ y_e & \text{otherwise} \end{cases}.
$$

Under the mapping, one can check that C-LP = C-LP$'$ and if the solution $x^{\text{C-LP}}$ of C-LP is unique and integral, the solution $y^{\text{C-LP}'}$ of C-LP$'$ is as well, i.e., $y^{\text{C-LP}'} = y^*$. Hence, (4) in Theorem 3.1 follows. Furthermore, since the solution $y^* = [y_e^*]$ to C-LP$'$ is unique and integral, there exists $c > 0$ such that

$$
c = \inf_{y \ne y^* : y \text{ is feasible to C-LP}'} \frac{w' \cdot (y^* - y)}{|y^* - y|},
$$

where $w' = [w'_e]$. Using this notation, we establish the following lemma characterizing performance of the max-product BP over the new GM (3). Theorem 3.1 follows from this lemma directly.

**Lemma 3.2.** *If the solution $y^{\text{C-LP}'}$ of C-LP$'$ is integral and unique, i.e., $y^{\text{C-LP}'} = y^*$, then*

- *If $y_e^* = 1$, $n_e^t[1] > n_e^t[0]$ for all $t > \frac{6 w'_{\max}}{c} + 6$,*

- *If $y_e^* = 0$, $n_e^t[1] < n_e^t[0]$ for all $t > \frac{6 w'_{\max}}{c} + 6$,*

*where $n_e^t[\cdot]$ denotes the BP belief of edge $e$ at time $t$ under the GM (3) and $w'_{\max} = \max_{e \in E'} |w'_e|$.*

## 3.2 Proof of Lemma 3.2

This section provides the complete proof of Lemma 3.2. We focus here on the case of $y_e^* = 1$, while translation of the result to the opposite case of $y_e^* = 0$ is straightforward. To derive a contradiction, assume that $n_e^t[1] \le n_e^t[0]$ and construct a tree-structured GM $T_e(t)$ of depth $t + 1$, also known as the computational tree, using the following scheme:

1. Add a copy of $Y_e \in \{0, 1\}$ as the (root) variable (with variable function $e^{w'_e Y_e}$).

2. Repeat the following $t$ times for each leaf variable $Y_e$ on the current tree-structured GM.

    2-1. For each $i \in V$ such that $e \in \delta(i)$ and $\psi_i$ is not associated to $Y_e$ of the current model, add $\psi_i$ as a factor (function) with copies of $\{Y_{e'} \in \{0,1\} : e' \in \delta(i) \setminus e\}$ as child variables (with corresponding variable functions, i.e., $\{e^{w'_{e'} Y_{e'}}\}$).

    2-2. For each $C \in \mathcal{C}$ such that $e \in \delta(i_C)$ and $\psi_C$ is not associated to $Y_e$ of the current model, add $\psi_C$ as a factor (function) with copies of $\{Y_{e'} \in \{0, 1\} : e' \in \delta(i_C) \setminus e\}$ as child variables (with corresponding variable functions, i.e., $\{e^{w'_{e'} Y_{e'}}\}$).

It is known from [17] that there exists a MAP configuration $y^{\text{TMAP}}$ on $T_e(t)$ with $y_e^{\text{TMAP}} = 0$ at the root variable. Now we construct a new assignment $y^{\text{NEW}}$ on the computational tree $T_e(t)$ as follows.

1. Initially, set $y^{\text{NEW}} \leftarrow y^{\text{TMAP}}$ and $e$ is the root of the tree.

2. $y^{\text{NEW}} \leftarrow \text{FLIP}_e(y^{\text{NEW}})$.

3. For each child factor $\psi$, which is equal to $\psi_i$ (i.e., $e \in \delta(i)$) or $\psi_C$ (i.e., $e \in \delta(i_C)$), associated with $e$,

    (a) If $\psi$ is satisfied by $y^{\text{NEW}}$ and $\text{FLIP}_e(y^*)$ (i.e., $\psi(y^{\text{NEW}}) = \psi(\text{FLIP}_e(y^*)) = 1$), then do nothing.

    (b) Else if there exists a $e$'s child $e'$ through factor $\psi$ such that $y_{e'}^{\text{NEW}} \neq y_{e'}^*$ and $\psi$ is satisfied by $\text{FLIP}_{e'}(y^{\text{NEW}})$ and $\text{FLIP}_{e'}(\text{FLIP}_e(y^*))$, then go to the step 2 with $e \leftarrow e'$.

    (c) Otherwise, report ERROR.

To aid readers understanding, we provide a figure describing an example of the above construction in our technical report [21]. In the construction, $\text{FLIP}_e(y)$ is the 0-1 vector made by flipping (i.e., changing from 0 to 1 or 1 to 0) the $e$'s position in $y$. We note that there exists exactly one child factor $\psi$ in step 3 and we only choose one child $e'$ in step (b) (even though there are many possible candidates). Due to this reason, flip operations induce a path structure $P$ in tree $T_e(t)$.[1] Now we state the following key lemma for the above construction of $y^{\text{NEW}}$.

**Lemma 3.3.** ERROR *is never reported in the construction described above.*

*Proof.* The case when $\psi = \psi_i$ at the step 3 is easy, and we only provide the proof for the case when $\psi = \psi_C$. We also assume that $y_e^{\text{NEW}}$ is flipped as $1 \to 0$ (i.e., $y_e^* = 0$), where the proof for the case $0 \to 1$ follows in a similar manner. First, one can observe that $y$ satisfies $\psi_C$ if and only if $y$ is the 0-1 indicator vector of a union of disjoint even paths in the cycle $C$. Since $y_e^{\text{NEW}}$ is flipped as $1 \to 0$, the even path including $e$ is broken into an even (possibly, empty) path and an odd (always, non-empty) path. We consider two cases: (a) there exists $e'$ within the odd path (i.e., $y_{e'}^{\text{NEW}} = 1$) such that $y_{e'}^* = 0$ and flipping $y_{e'}^{\text{NEW}}$ as $1 \to 0$ broke the odd path into two even (disjoint) paths; (b) there exists no such $e'$ within the odd path.

For the first case (a), it is easy to see that we can maintain the structure of disjoint even paths in $y^{\text{NEW}}$ after flipping $y_{e'}^{\text{NEW}}$ as $1 \to 0$, i.e., $\psi$ is satisfied by $\text{FLIP}_{e'}(y^{\text{NEW}})$. For the second case (b), we choose $e'$ as a neighbor of the farthest end point (from $e$) in the odd path, i.e., $y_{e'}^{\text{NEW}} = 0$ (before flipping). Then, $y_{e'}^* = 1$ since $y^*$ satisfies factor $\psi_C$ and induces a union of disjoint even paths in the cycle $C$. Therefore, if we flip $y_{e'}^{\text{NEW}}$ as $0 \to 1$, then we can still maintain the structure of disjoint even paths in $y^{\text{NEW}}$, $\psi$ is satisfied by $\text{FLIP}_{e'}(y^{\text{NEW}})$. The proof for the case of the $\psi$ satisfied by $\text{FLIP}_{e'}(\text{FLIP}_e(y^*))$ is similar. This completes the proof of Lemma 3.3. $\square$

Due to how it is constructed $y^{\text{NEW}}$ is a valid configuration, i.e., it satisfies all the factor functions in $T_e(t)$. Hence, it suffices to prove that $w'(y^{\text{NEW}}) > w'(y^{\text{TMAP}})$, which contradicts to the assumption that $y^{MAP}$ is a MAP configuration on $T_e(t)$. To this end, for $(i, j) \in E'$, let $n_{ij}^{0 \to 1}$ and $n_{ij}^{1 \to 0}$ be the number of flip operations $0 \to 1$ and $1 \to 0$ for copies of $(i, j)$ in the step 2 of the construction of $T_e(t)$. Then, one derives

$$w'(y^{\text{NEW}}) = w'(y^{\text{TMAP}}) + w' \cdot n^{0 \to 1} - w' \cdot n^{1 \to 0},$$

where $n^{0 \to 1} = [n_{ij}^{0 \to 1}]$ and $n^{1 \to 0} = [n_{ij}^{1 \to 0}]$. We consider two cases: (i) the path $P$ does not arrive at a leave variable of $T_e(t)$, and (ii) otherwise. Note that the case (i) is possible only when the condition in the step (a) holds during the construction of $y^{\text{NEW}}$.

**Case (i).** In this case, we define $y_{ij}^\dagger := y_{ij}^* + \varepsilon(n_{ij}^{1 \to 0} - n_{ij}^{0 \to 1})$, and establish the following lemma.

**Lemma 3.4.** $y^\dagger$ *is feasible to C-LP$'$ for small enough $\varepsilon > 0$.*

*Proof.* We have to show that $y^\dagger$ satisfies (5) and (6). Here, we prove that $y^\dagger$ satisfies (6) for small enough $\varepsilon > 0$, and the proof for (5) can be argued in a similar manner. To this end, for given $C \in \mathcal{C}$,

we consider the following polytope $\mathcal{P}_C$ :

$$\sum_{j \in V(C)} y_{i_C,j} \le |C| - 1, \quad y_{i_C,j} \in [0,1], \quad \forall j \in C, \quad \sum_{j \in V(C)} (-1)^{d_C(j,e)} y_{i_C,j} \in [0,2], \quad \forall e \in E(C).$$

We have to show that $y_C^{\dagger} = [y_e : e \in \delta(i_C)]$ is within the polytope. It is easy to see that the condition of the step (a) never holds if $\psi = \psi_C$ in the step 3. For the $i$-th copy of $\psi_C$ in $P \cap T_e(t)$, we set $y_C^*(i) = \mathtt{FLIP}_{e'}(\mathtt{FLIP}_e(y_C^*))$ in the step (b), where $y_C^*(i) \in \mathcal{P}_C$. Since the path $P$ does not hit a leave variable of $T_e(t)$, we have

$$\frac{1}{N} \sum_{i=1}^{N} y_C^*(i) = y_C^* + \frac{1}{N} \left( n_C^{1 \to 0} - n_C^{0 \to 1} \right),$$

where $N$ is the number of copies of $\psi_C$ in $P \cap T_e(t)$. Furthermore, $\frac{1}{N} \sum_{i=1}^{N} y_C^*(i) \in \mathcal{P}_C$ due to $y_C^*(i) \in \mathcal{P}_C$. Therefore, $y_C^{\dagger} \in \mathcal{P}_C$ if $\varepsilon \le 1/N$. This completes the proof of Lemma 3.4. $\qquad\square$

The above lemma with $w'(y^*) > w'(y^{\dagger})$ (due to the uniqueness of $y^*$) implies that $w' \cdot n^{0 \to 1} > w' \cdot n^{1 \to 0}$, which leads to $w'(y^{\mathrm{NEW}}) > w'(y^{\mathrm{TMAP}})$.

**Case (ii).** We consider the case when only one end of $P$ hits a leave variable $Y_e$ of $T_e(t)$, where the proof of the other case follows in a similar manner. In this case, we define $y_{ij}^{\ddagger} := y_{ij}^* + \varepsilon(m_{ij}^{1 \to 0} - m_{ij}^{0 \to 1})$, where $m^{1 \to 0} = [m_{ij}^{1 \to 0}]$ and $m^{0 \to 1} = [m_{ij}^{0 \to 1}]$ is constructed as follows:

---

1. Initially, set $m^{1 \to 0}, m^{0 \to 1}$ by $n^{1 \to 0}, n^{0 \to 1}$.

2. If $y_e^{\mathrm{NEW}}$ is flipped as $1 \to 0$ and it is associated to a cycle parent factor $\psi_C$ for some $C \in \mathcal{C}$, then decrease $m_e^{1 \to 0}$ by 1 and

    2-1 If the parent $y_{e'}^{\mathrm{NEW}}$ is flipped from $1 \to 0$, then decrease $m_{e'}^{1 \to 0}$ by 1.

    2-2 Else if there exists a 'brother' edge $e'' \in \delta(i_C)$ of $e$ such that $y_{e''}^* = 1$ and $\psi_C$ is satisfied by $\mathtt{FLIP}_{e''}(\mathtt{FLIP}_{e'}(y^*))$, then increase $m_{e''}^{0 \to 1}$ by 1.

    2-3 Otherwise, report ERROR.

3. If $y_e^{\mathrm{NEW}}$ is flipped as $1 \to 0$ and it is associated to a vertex parent factor $\psi_i$ for some $i \in V$, then decrease $m_e^{1 \to 0}$ by 1.

4. If $y_e^{\mathrm{NEW}}$ is flipped as $0 \to 1$ and it is associated to a vertex parent factor $\psi_i$ for some $i \in V$, then decrease $m_e^{0 \to 1}, m_{e'}^{1 \to 0}$ by 1, where $e' \in \delta(i)$ is the 'parent' edge of $e$, and

    4-1 If the parent $y_{e'}^{\mathrm{NEW}}$ is associated to a cycle parent factor $\psi_C$,

        4-1-1 If the grad-parent $y_{e''}^{\mathrm{NEW}}$ is flipped from $1 \to 0$, then decrease $m_{e''}^{1 \to 0}$ by 1.

        4-1-2 Else if there exists a 'brother' edge $e''' \in \delta(i_C)$ of $e'$ such that $y_{e'''}^* = 1$ and $\psi_C$ is satisfied by $\mathtt{FLIP}_{e'''}(\mathtt{FLIP}_{e''}(y^*))$, then increase $m_{e'''}^{0 \to 1}$ by 1.

        4-1-3 Otherwise, report ERROR.

    4-2 Otherwise, do nothing.

---

We establish the following lemmas.

**Lemma 3.5.** ERROR *is never reported in the above construction.*

**Lemma 3.6.** $y^{\ddagger}$ *is feasible to C-LP$'$ for small enough $\varepsilon > 0$.*

Proofs of Lemma 3.5 and Lemma 3.6 are analogous to those of Lemma 3.3 and Lemma 3.4, respectively. From Lemma 3.6, we have

$$c \le \frac{w' \cdot (y^* - y^{\ddagger})}{|y^* - y^{\ddagger}|} \le \frac{\varepsilon \left( w'(m^{0 \to 1} - m^{1 \to 0}) \right)}{\varepsilon(t-3)} \le \frac{\varepsilon \left( w'(n^{0 \to 1} - n^{1 \to 0}) + 3w'_{\max} \right)}{\varepsilon(t-3)},$$

where $|y^* - y^{\ddagger}| \ge \varepsilon(t-3)$ follows from the fact that $P$ hits a leave variable of $T_e(t)$ and there are at most three increases or decreases in $m^{0 \to 1}$ and $m^{1 \to 0}$ in the above construction. Hence,

$$w'(n^{0 \to 1} - n^{1 \to 0}) \ge c(t-3) - 3w'_{\max} > 0 \qquad \text{if} \quad t > \frac{3w'_{\max}}{c} + 3,$$

which implies $w'(y^{\text{NEW}}) > w'(y^{\text{TMAP}})$. If both ends of $P$ hit leave variables of $T_e(t)$, we need $t > \frac{6w'_{\max}}{c} + 6$. This completes the proof of Lemma 3.2.

## 4 Cutting-Plane Algorithm using Belief Propagation

In the previous section we established that BP on a carefully designed GM using non-intersecting odd-sized cycles solves the MWM problem when the corresponding MWM-LP relaxation is tight. However, finding a collection of odd-sized cycles to ensure tightness of the MWM-LP is a challenging task. In this section, we provide a heuristic algorithm which we call CP-BP (cutting-plane using BP) for this task. It consists of making sequential, "cutting plane", modifications to the underlying LP (and corresponding GM) using the output of the BP algorithm in the previous step. CP-BP is defined as follows:

1. Initialize $\mathcal{C} = \emptyset$.
2. Run BP on the GM in (3) for $T$ iterations
3. For each edge $e \in E$, set $y_e = \begin{cases} 1 & \text{if } n_e^T[1] > n_e^T[0] \text{ and } n_e^{T-1}[1] > n_e^{T-1}[0] \\ 0 & \text{if } n_e^T[1] < n_e^T[0] \text{ and } n_e^{T-1}[1] < n_e^{T-1}[0] \\ 1/2 & \text{otherwise} \end{cases}$.
4. Compute $x = [x_e]$ using $y = [y_e]$ as per (4), and terminate if $x \notin \{0, 1/2, 1\}^{|E|}$.
5. If there is no edge $e$ with $x_e = 1/2$, return $x$. Otherwise, add a non-intersecting odd-sized cycle of edges $\{e : x_e = 1/2\}$ to $\mathcal{C}$ and go to step 2; or terminate if no such cycle exists.

In the above procedure, BP can be replaced by an LP solver to directly obtain $x$ in step 4. This results in a traditional cutting-plane LP (CP-LP) method for the MWM problem [20]. The primary reason why we design CP-BP to terminate when $x \notin \{0, 1/2, 1\}^{|E|}$ is because the solution $x$ of C-LP is always half integral [2]. Note that $x \notin \{0, 1/2, 1\}^{|E|}$ occurs when BP fails to find the solution to the current MWM-LP.

We compare CP-BP and CP-LP in order to gauge the effectiveness of BP as an LP solver for MWM problems. We conducted experiments on two types of synthetically generated problems: 1) Sparse Graph instances; and 2) Triangulation instances. The sparse graph instances were generated by forming a complete graph on $|V| = \{50, 100\}$ nodes and independently eliminating edges with probability $p = \{0.5, 0.9\}$. Integral weights, drawn uniformly in $[1, 2^{20}]$, are assigned to the remaining edges. The triangulation instances were generated by randomly placing $|V| = \{100, 200\}$ points in the $2^{20} \times 2^{20}$ square and computing a Delaunay triangulation on this set of points. Edge weights were set to the rounded Euclidean distance between two points. A set of 100 instances were generated for each setting of $|V|$ and CP-BP was run for $T = 100$ iterations.

The results are summarized in Table 1 and show that: 1) CP-BP is almost as good as CP-LP for solving the MWM problem; and 2) our graphical transformation allows BP to solve significantly more MWM problems than are solvable by BP run on the 'bare' LP without odd-sized cycles.

| 50 % sparse graphs | | | | 90 % sparse graphs | | | |
|---|---|---|---|---|---|---|---|
| $|V| / |E|$ | # CP-BP | # Tight LPs | # CP-LP | $|V| / |E|$ | # CP-BP | # Tight LPs | # CP-LP |
| 50 / 490 | 94 % | 65 % | 98 % | 50 / 121 | 90 % | 59 % | 91 % |
| 100 / 1963 | 92 % | 48 % | 95 % | 100 / 476 | 63 % | 50 % | 63 % |

| | Triangulation, $|V| = 100, |E| = 285$ | | Triangulation, $|V| = 200, |E| = 583$ | |
|---|---|---|---|---|
| Algorithm | # Correct / # Converged | Time (sec) | # Correct / # Converged | Time (sec) |
| CP-BP | 33 / 36 | 0.2 [0.0,0.4] | 11 / 12 | 0.9 [0.2,2.5] |
| CP-LP | 34 / 100 | 0.1 [0.0,0.3] | 15 / 100 | 0.8 [0.3,1.6] |

Table 1: Evaluation of CP-BP and CP-LP on random MWM instances. Columns *# CP-BP* and *# CP-LP* indicate the percentage of instances in which the cutting plane methods found a MWM. The column *# Tight LPs* indicates the percentage for which the initial MWM-LP is tight (i.e. $\mathcal{C} = \emptyset$). *# Correct* and *# Converged* indicate the number of correct matchings and number of instances in which CP-BP converged upon termination, but we failed to find a non-intersecting odd-sized cycle. The *Time* column indicates the *mean [min,max]* time.

## Footnotes

*Also at Theoretical Division of Los Alamos National Lab.

[1] $P$ may not have an alternating structure since both $y_e^{\text{NEW}}$ and its child $y_{e'}^{\text{NEW}}$ can be flipped in a same way.

[2] A proof of $\frac{1}{2}$-integrality, which we did not find in the literature, is presented in our technical report [21].

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
