[Reviews · NeurIPS 2013]

Submitted by Assigned_Reviewer_5

I've read the reviewers' feedback. I like the overall summary. It would be good to emphasize this perspective more clearly in the paper.

The paper introduces a BP algorithm for finding a Max Weight Matching (MWM) in a general weighted graph. Convergence is proved provided the LP relaxation is tight. Key contribution involves a transformation of the underlying GM to deal with odd cycles and force convergence. The result leads to a novel BP-based heuristic CP-BP (cutting plane method using BP) for the MWM problem. Experiments show the algorithm to be competitive with a CP-LP (cutting plane method using LP) approach. The graphical transformation introduced allows BP to solve many more instances than running BP on the original graph instances.

Solid, novel contribution, extending the growing body of work building connections between BP and LP and leading to a better understanding of BP on loopy GMs.

I did not check all the details but the analysis appears correct.

One issue that I would like to see clarified a bit further concerns the CP-BP algorithm itself. The algorithm is based on the main result from the paper that shows that "BP on a carefully designed GM using appropriate odd cycles solves the MWM problem *as long as the corresponding MWM-LP relaxation is tight.* When you return x in step 5 of CP-BP, are you guaranteed that this x is optimal? (i.e., do you know at that poin that the relaxation is tight?) In a result by Sanghavi (IEEE 2007) the connection between LP and BP is shown in both directions: (1) LP tight ===> max product BP converges and to the correct answer, (2) LP is loose ===> BP does not converge. I did not see that second component in this paper. I may simply have missed it.

My only reservation about the result concerns the question of how broad the appeal of the new result is. But, in any case, it's definitely a solid step forward in the sequence of papers studying connections between LP and BP and the convergences of BP in loopy settings.
Summary: Paper introduces a BP approach to finding Maximum Weight Matchings using a transformation of the original graph structure. Experimental evidence shows the method competitive with an LP based approach.

Submitted by Assigned_Reviewer_6

The authors extend a known theorem that states that, if the linear relaxation of an MWM problem is tight, then BP will converge correctly on the corresponding graphical model. While this previously was only shown with no "blossom" inequalities -- i.e. inequalities that help to achieve integrality of the LP -- the paper extends this theorem to hold also with such inequalities. That allows more instances of MWM to be solved via BP.

This reviewer thinks that the paper adds a tiny puzzle piece to understand loopy BP properties, but is not perfectly clear in the presentation and lacks an (experimental) proof of the usefulness of the results.

Clarity of presentation:
- The is \Omega a set of node indices or the value range of Z_i?
- The GM is introduced in L104 without ever referring to a graph. ?
- The notation in the introduction is not very clear, which conditions are necessary and which are sufficient (L50: "BP converges ... when LP tight" ==> supposed to mean not sufficient, L68 "BP converges ... if LP is tight" ==> supposed to mean sufficient).
- What is an odd-sized cycle? |C| odd or |E(C)| odd?
- L139: For general (non-pairwise-potential) GM problems, the messages could have more than one free variable.
- L179: What is V(C)?
- "it is not hard to check ... updates involving \Phi_C can be done using dynamic programming" I cannot see that.
- Figure 1 is not intuitive and is not explained sufficiently.
- L293-306 The proof's correctness is obstructed by too complicated notation (e.g. why is y_{e’}{NEW}=1? Is this implied by the odd/even path in your notation?)

Usefulness of the results:
- Why not simply do LP? The only real advantage to me seems that BP can be implemented in a distributed fashion. But if so, can it also be checked in a distributed fashion, whether the LP is tight (and thus the BP result is trustworthy)?
- The experiments should not only use random graphs but real ones. Otherwise the benefit of the blossom inequalities may be overrated.
- The experiments should show beneficial timings or some other proof of the advantage of the proposed method.
- The potentials \Phi_C potentially contain many variables. Summing over them during the BP update may become exponentially hard.

Response to the authors' rebuttal: In their rebuttal, the authors have put their work in the light of "making LP ready for parallel computation". Moreover, they have promised to add more experimental results to support this claim. I think this line of argumentation adds an interesting perspective to the work. It should be elaborated on in the final version of the paper.
Summary: A modified BP algorithm to solve a subset of MWM problems (how large?) in a distributed fashion.

Submitted by Assigned_Reviewer_7

Summary: The paper extends the analysis of belief propagation (BP) by Sanghavi et al. [6] for the problem of maximum weight matching (MWM). The assumption in [6] was that the graph contains no odd cycles. Under this assumption, if the LP for the MWM problem was tight and had a unique solution, [6] proved that BP provides the correct matching. In this paper, an odd cycle is handled by transforming the underlying graphical model for BP, specifically, by replacing the odd cycle with a new vertex that is connected to all the vertices of the original cycle. The paper also presents a heuristic cutting plane style algorithm which identifies an odd cycle in the graph and replaces it with a new node iteratively (denoted by CP-BP). Experiments on a synthetic dataset show that the proposed algorithm provides similar accuracy to a cutting plane LP algorithm (denoted CP-LP).

Quality: The paper appears to be technically correct, but I did not check the proofs thoroughly. The choice of the baseline is not well motivated. Why not evaluate with respect to the cutting plane LP algorithm of [13], which provably provides the optimal matching in polynomial time? Or with any of the other specialized softwares for solving the maximum weighted matching problem (e.g. Blossom V by Vladimir Kolmogorov)?

Clarity: The paper is mostly clear. However, in my opinion, the introduction does not provide the right motivation for this work. It is claimed that a BP based approach for MWM would be more efficient than LP solvers. While this may be true if we use a standard interior-point or simplex solver (as suggested in the introduction), MWM has a long history in the combinatorial optimization literature. There are several specialized solvers for this problem that are very efficient in practice. Recently, there has also been a lot of work to parallelize LP solvers.

I think the right motivation for the work is to gain a better understanding of BP, which is a classic algorithm used in the machine learning community.

If, however, the motivation really was to design efficient solvers for MWM, then more thorough experiments should be performed, with real datasets, and the strongest possible LP baselines available.

Originality: To the best of my knowledge, this is a novel extension of the analysis in [6].

Significance: In my opinion, this paper will not impact research practices. There is no evidence that the heuristic CP-BP approach is more efficient than the state of the art solvers. However, the analysis should be of interest to a subset of the NIPS audience.
Summary: Theoretically interesting paper that presents an analysis of a classical inference algorithm, but limited practical use.

Post-rebuttal comments: The additional experiments that the rebuttal promises will certainly help improve the quality of the paper. However, without seeing the results, it is difficult for me to change my score for the paper.

Regarding parallelization, it is not immediately obvious to me why BP is more suited to parallelization than LP based methods since a lot of effort has been put in to make convex programming parallel via decomposition methods (ADMM, Lagrangian relaxation etc.). If there is indeed a strong argument for BP being a better candidate for parallelization, it would make a very interesting discussion topic. So I hope that subsequent versions of the paper highlight this.
Author Feedback

Author rebuttal: First of all, we very much appreciate the valuable comments, effort and time of the reviewers on this paper. In what follows, we provide a summary response for all reviewers, followed by detailed response to each reviewer.

Summary for all reviewers:
Most prior work on BP and LP has focused merely on BP "analysis." To our knowledge, we are the first to suggest how to "fix" BP via a graph transformation so that it works properly, i.e., recovers the desired LP solution. This is the novel characteristic of our work and we think it is an important/solid contribution for the BP community, as does Reviewer 5. We also believe our work is of broad interest to the parallel machine learning audience in NIPS, due to the many real-world applications of MWM and recent parallel implementations of BP, e.g. via MapReduce [Gonzalez et al. JMLR 2009] & GraphLab [Low et al. UAI 2010]. Following Reviewer 6 & 7's suggestions, we will include additional experiments in the camera-ready version to further validate our work (beyond theory) and also compare to other known algorithms.


For Reviewer 5:
In our experiments, we found that on a GM with odd cycles, BP often converges even when the corresponding LP is loose (but it very rarely occurs and we only found one or two such MWM instances). This means that an analogous version (i.e. LP is loose ===> BP does not converge) of what [Sanghavi et al. IEEE 2007] found in the standard GM without odd cycles no longer holds in our more general GM with odd cycles. (However, odd cycles can tighten LP even though BP may not check it.) Furthermore, in our experiments, we observe that our algorithm considering odd cycles outperforms that of [Sanghavi et al. IEEE 2007] (irrespective of the tightness of LP). In the current draft, we provide partial experimental results for this (due to the limited space), but will add more ones to the camera-ready version (e.g. including what happens when LP is loose) by referring some parts of our proof to a technical report. We appreciate again Reviewer 5's valuable comments/questions.


For Reviewer 6:
We will incorporate the following in the camera-ready version. We appreciate Reviewer 6's valuable feedback.

- \Omega is the value range of Z_i, i.e., \Omega={0,1} in our case.
- We first introduce general formulation of GM without referring to a graph and later describe specific GMs dependent on given graph and edge weights.
- In the introduction, we mean "sufficient" by "when LP is tight". We will revise it to "if LP is tight" following Reviewer 6's suggestion.
- The odd cycle has an odd number of edges or vertices, i.e., |E(C)|=|V(C)| is odd.
- V(C) is the set of vertices of odd cycle C.
- Message updates involving \Phi_C may become exponentially hard as Reviewer 6 pointed out. However, this can be done efficiently (in linear time) using dynamic programming, as we mentioned briefly in the draft. We will explain more details about the dynamic programming in the camera-ready version.
- We will update Figure 1 with more intuitive one and explanations, as Reviewer 6 pointed out.
- We will also revise the proof with much simplified notation.
- About whether the LP tightness is checkable in a distributed fashion, we refer our earlier response to Reviewer 5's comment.
- As we mentioned in the summary, we will add more experiments (with real graphs) to justify the validity of our work comparing with existing algorithms (including LP-based ones).
- As we mentioned in the summary, our two major contributions are the followings. We are first to suggest how to fix BP via a graph transformation so that it recovers the desired LP solution - which is a novel contribution in our work, not existing in prior BP analytic works. Furthermore, BP can be implementable in a distributed/parallel fashion, which is an advantage of our BP approach comparing with LP-based ones. I think this second aspect is of broader interest to the parallel machine learning audience in NIPS.


For Reviewer 7:
We will incorporate the following in the camera-ready version. We appreciate Reviewer 7's valuable feedback.

- Our algorithm is significantly simpler to implement than the LP-based cutting plain algorithm [13], while both guarantee poly-time complexity in theory (our algorithm additionally requires LP tightness). In particular, our algorithm runs in O(n^3) time (it is not hard to check this for O(1) weights using Lemma 3.2 but we will add formal justification for this in the camera-ready version). On the other hand, [13] runs in \Omega(n^5) time (i.e. the best known complexity of generic LP solver is \Omega(n^4) and [13] requires to solve \Omega(n) LPs). Nevertheless, as we mentioned earlier in the summary, we will add more experimental results with real datasets to the camera-ready version comparing our BP algorithm with strongest known algorithms including the LP-based cutting plane algorithm [13] and the Blossom V by Kolmogorov.
- As Reviewer 7 mentioned, MWM has a long history. However, most existing algorithms are centralized and are hard to implement in a distributed/parallel fashion. BP is a strong candidate as a parallel solver for generic MWM and our work provides a solid contribution on this line. Furthermore, although there has been much effort to parallelize LP solvers, it is still an important open question (to our knowledge) and we think BP can is a strong candidate for a parallel LP solver. For these reasons as well as numerous real-world applications of the MWM problem, we believe that our work is of broader interest to the parallel machine learning audience in NIPS.